# Learning Vortex Enhancement with Angular-Speed-Invariant Importance Sampling in SPH Fluids

## Abstract

Learning vortex enhancement in SPH benefits most from what is sampled. We use angular-speed-invariant importance sampling with the Kinematic Vorticity Number (KVN) to target vortex cores across resolutions and flow speeds. Particles selected by KVN are pooled into a lightweight global token via attention. Models are trained with velocity correction targets obtained by applying a Biot–Savart mapping to the vorticity loss field. Compared with uniform and vorticity-based sampling, KVN-based sampling improves vortex coherence and advances the emergence of secondary vortices across scenes and particle counts. The gains persist under coarse and fine discretizations and scale smoothly with particle count, indicating robustness to resolution changes. Ablations further show that injecting KVN-based information also benefits alternative encoder variants, suggesting that angular-speed-invariant sampling is a simple, transferable lever for learning vortex enhancement in SPH.

## 1 Introduction

Simulating visually rich flows remains a central goal in computer graphics. Smoothed Particle Hydrodynamics (SPH) is a widely used Lagrangian formulation in which fluids are discretized as particles that move with the velocity field (Wang et al. (2024)). Despite avoiding explicit advection, frequent kernel summations and coarse discretization introduce numerical dissipation that smooths high-frequency structures such as vortices (Koschier et al. (2019)). Classical remedies inject rotation using vorticity-based forces (Fedkiw et al. (2001)) or micro-rotations (Bender et al. (2019)). Learning-based enhancement has emerged as an alternative (Jain et al. (2024)). A central question in these pipelines is which particles to sample or attend to. In practice, many works use vorticity magnitude as the importance signal (Rioux-Lavoie et al. (2022)). However, vorticity scales with angular speed (Bridson (2015)), which biases models toward already fast-spinning structures and overlooks nascent, low-speed vortices (Ye et al. (2025)).

We address this limitation with an angular-speed-invariant importance signal based on the Kinematic Vorticity Number (KVN), a dimensionless measure of how closely local deformation resembles rigid-body rotation, independent of rotation rate (Schielicke et al. (2016)). High KVN particles are sampled and pooled via attention into a lightweight global token that targets vortex cores across resolutions and flow speeds. Models are trained with velocity-correction targets obtained by applying the Biot–Savart operator to the vorticity loss, providing a physics-based supervision signal. We evaluate sampling strategies across scenes and particle counts. Compared with uniform and vorticity-based sampling, KVN-based sampling improves vortex coherence and advances the emergence of secondary vortices under multiple spatial discretizations.

Our contributions are summarized as: (i) an angular-speed-invariant importance signal for learning vortex enhancement in SPH instantiated with KVN; (ii) a systematic study showing cross-resolution robustness and a clear sample-budget accuracy curve; and (iii) evidence that KVN-based information improves alternative encoder configurations under equal capacity and training.

## 2 RELATED WORK

**Learning Models for Fluid Simulations.** Learning high-fidelity fluid details has been approached for many years. One stream performs data-driven super-resolution or multi-resolution training (Bai et al. (2021; 2020); Chu & Thuerey (2017)). Recent work focuses on understanding fluid structure to forecast future states by employing Convolutional Neural Networks (CNN) (Xiao et al. (2020)), Graph Neural Networks (GNN) (Sanchez-Gonzalez et al. (2020); Toshev et al. (2024)), transformers (Shao et al. (2022)), or in hybrid forms (Janny et al. (2023); Xu & Li (2024)). Several models target specific phenomena such as splashes (Um et al. (2018)) and vortices (Xiong et al. (2023); Deng et al. (2023a)), or pursue implicit/reduced dynamics (Tao et al. (2024)). In parallel, some work brings Physics-Informed Neural Networks (PINNs) to fluid field fitting (Lee et al. (2025); Chen et al. (2025)), integrates differentiable solvers into backward propagation (Um et al. (2020); Tathawadekar et al. (2023)), and also combines with advanced numerical approaches such as flow maps (Deng et al. (2023b)) or the Monte Carlo PDE solvers (Jain et al. (2024)). In Lagrangian settings, the typical task is to learn particle-wise mappings along continuous time sequences (Ummenhofer et al. (2020); Zhang et al. (2020)). Ma et al. further track adaptively sampled key particles to guide predictions (Ma et al. (2024)). A critical design choice in learning-based vortex enhancement is the importance sampling strategy for selecting informative particles. Our work demonstrates that angular-speed-invariant sampling using the KVN outperforms vorticity-based importance measures.

**Point Cloud Encoding.** Point-cloud networks are a natural fit for Lagrangian fluids. Early work focuses on local set-abstraction and neighborhood operators with shared MLPs and symmetric pooling (Charles et al. (2017)). This architecture is upgraded by hierarchical sampling and grouping (Qi et al. (2017)) and dynamic or kernelized neighborhoods (Wang et al. (2019); Wu et al. (2019); Thomas et al. (2019); Xu et al. (2021)). Attention and pretraining are also introduced to represent global context with full or hierarchical transformers (Zhao et al. (2021); Wu et al. (2022); Lai et al. (2022); Lee et al. (2019)). Meanwhile, masked point pretraining and strong baselines further advance performance (Yu et al. (2022); Pang et al. (2022); Qian et al. (2022); Wang et al. (2025)). Representative point selection sometimes decouples grouping radii from feature learning (Dovrat et al. (2019); Kool et al. (2020)). Symmetric Fourier-basis continuous convolutions demonstrate strong accuracy and generalization on SPH fluids (Winchenbach & Thuerey (2024)). Our approach combines continuous convolutions for local encoding with physics-guided importance sampling for global context extraction.

## 3 PRELIMINARIES

**SPH Simulation Pipeline.** SPH simulates incompressible fluids by tracking discrete particles that evolve under forces such as pressure, viscosity, and external fields. The particle positions $\mathbf{x}$ and velocities $\mathbf{u}$ are updated via the Lagrangian form of the Navier–Stokes equation. At each simulation step, key operations include: (i) computing densities and pressure forces via kernel-weighted summations over neighbors; (ii) solving incompressibility; and (iii) updating particle positions and velocities. However, due to frequent local averaging and coarse discretization (Koschier et al. (2019)), SPH tends to dissipate high-frequency structures like vortices.

**Velocity Correction for Vortices in SPH.** To characterize the decay of rotational features, we adopt a formulation of vorticity loss $\delta\boldsymbol{\omega}$, defined as the discrepancy between the evolved vorticity and the intermediate velocity field:

$$\delta\boldsymbol{\omega} = \boldsymbol{\omega}^{n+1} - \nabla \times \bar{\mathbf{u}}, \tag{1}$$

where $\bar{\mathbf{u}}$ refers to the velocity after applying non-pressure forces and $\boldsymbol{\omega} = \nabla \times \mathbf{u}$ denotes the vorticity. As shown in prior work (Zhang et al. (2015)), compensating for this vorticity loss using velocity corrections derived from the Biot–Savart law can effectively reinforce rotational features. This quantity captures dissipated vortex structures not restored by standard SPH operations. Following Ye et al. (2025), we convert the vorticity loss $\delta\boldsymbol{\omega}$ into a velocity correction $\delta\mathbf{u}$ using a Biot–Savart convolution:

$$\delta\mathbf{u}_i = \frac{v}{|\mathcal{S}|} \sum_{j \in \mathcal{S}} \frac{\nabla G(\mathbf{x}_i - \mathbf{x}_j) \times \delta\boldsymbol{\omega}_j}{P_j}, \tag{2}$$

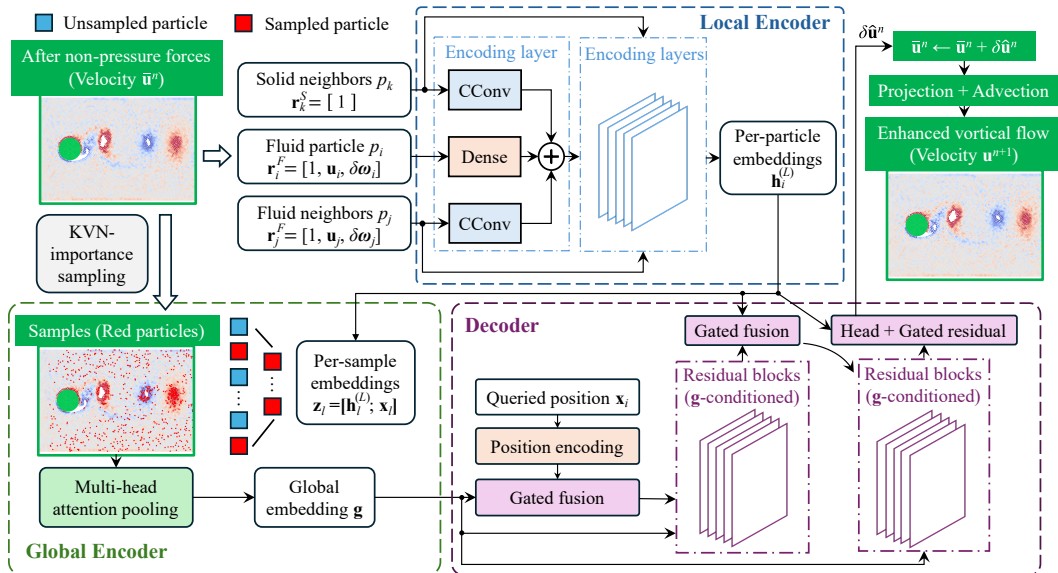

Figure 1: Our method combines local continuous convolutions with KVN-based global tokenization to enhance vortical structures in SPH fluids. The local encoder processes fluid and solid particles through fixed-radius continuous convolutions, capturing detailed neighborhood interactions. KVN-based importance sampling selects representative particles (red) from the fluid domain, which are then aggregated into a global token via pooled multi-head attention. The decoder fuses global context with local embeddings through a two-stage conditional architecture, predicting velocity corrections $\delta\hat{\mathbf{u}}^n$ that enhance vortical flow while preserving physical consistency.

where $\mathcal{S}$ is a sample set, $P_j$ is the sampling probability, $v$ is the volume of a vortex particle, and $G$ is the Green's function. This velocity correction serves as the supervision target for training. The process computing the supervision target is detailed in Appendix. B.

## 4 LEARNING FRAMEWORK WITH ANGULAR-SPEED-INVARIANT SAMPLING

We present a learning framework that addresses the fundamental challenge of capturing vortical dynamics through angular-speed-invariant importance sampling. Our approach combines KVN-based global tokenization with local continuous convolutions to enhance vortical structures in SPH simulations. Figure 1 illustrates the overall architecture, which consists of three main components: a local encoder for high-fidelity neighborhood aggregation, a global encoder for physics-guided context modeling using the Kinematic Vorticity Number (KVN), and a decoder that fuses local and global information to predict velocity corrections.

### 4.1 LOCAL ENCODER

The local encoder captures fine-grained fluid dynamics through Continuous Convolutions (CConv) over particle neighborhoods (Ummenhofer et al. (2020)). We employ fixed radius continuous convolutions to preserve the physical continuity inherent in fluid systems. For each particle $p_i$, we construct feature vectors that incorporate both kinematic and vortical properties $\mathbf{r}_i^F = [1, \mathbf{u}_i, \delta\boldsymbol{\omega}_i]$, where $\mathbf{u}_i$ is the velocity, $\delta\boldsymbol{\omega}_i$ is the vorticity loss. The leading 1 serves as a constant feature channel that provides a reference magnitude for the continuous convolution operations. The local encoder processes both fluid and solid particles within a fixed radius of each queried particle. Solid particles are represented with a simplified feature vector $\mathbf{r}^S = [1]$, as they primarily provide boundary context. The encoder consists of multiple layers. Continuous convolutions aggregate features from fluid and solid neighbors by

$$\mathbf{h}_i^{(l+1)} = \text{CConv}_{\text{fluid}}(\{\mathbf{r}_j^F\}_{j \in \mathcal{N}_i^F}) + \text{CConv}_{\text{solid}}(\{\mathbf{r}_k^S\}_{k \in \mathcal{N}_i^S}) + \text{Dense}(\mathbf{h}_i^{(l)}), \tag{3}$$

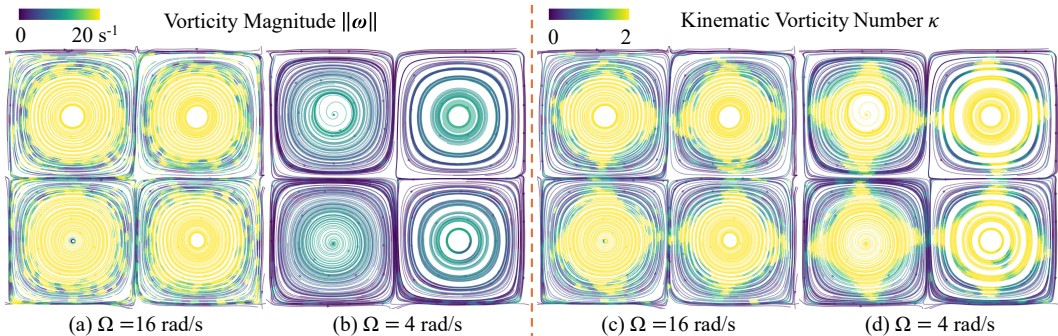

Figure 2: Vorticity inherently scales with angular speed $\Omega$, resulting in less attention to low-angular-speed nascent vortices. The angular-speed-invariant KVN provides equal measurement of vortices with various angular speeds.

where $\mathbf{h}_i^{(l)}$ denotes local embeddings after the $l^{th}$ encoding layer, $\mathcal{N}_i^F$ and $\mathcal{N}_i^S$ refer to the fluid and solid neighbors of particle $p_i$, respectively. Dense layers subsequently transform particle representations within the same spatial location. The continuous convolution employs a Poly6 window function to ensure smooth spatial weighting as

$$w(\mathbf{x}_i, \mathbf{x}_j) = \max(0, (1 - \frac{\|\mathbf{x}_i - \mathbf{x}_j\|^2}{r_{\max}^2})^3), \tag{4}$$

where $r_{\max}$ denotes the maximum radius for searching neighbors. This design allows the local encoder to capture detailed fluid-solid interactions while maintaining computational efficiency through fixed-radius neighborhoods.

## 4.2 KVN-BASED SAMPLING AND GLOBAL CONTEXT EXTRACTION

Effective importance sampling for vortex enhancement requires identifying rotational structures based on their geometric properties rather than instantaneous speeds. While vorticity magnitude is commonly used to locate vortex-rich regions, it inherently scales with angular speed, thus favoring fast-rotating vortices and overlooking nascent structures as shown in Fig. 2(a) and (b).

We instead adopt the Kinematic Vorticity Number (KVN), a dimensionless and angular-speed-invariant metric (see Fig. 2(c) and (d)) defined as

$$\kappa = \frac{\|\boldsymbol{\omega}\|}{2\|\mathbf{S}\|}, \quad \text{where } \mathbf{S} = \frac{1}{2}(\nabla \mathbf{u} + \nabla \mathbf{u}^\top), \tag{5}$$

where $\mathbf{S}$ denotes the strain tensor. High $\kappa$ values indicate rigid-body-like rotational regions, making KVN more suitable for identifying vortex cores irrespective of rotation speed. This angular-speed invariance enables our method to identify nascent low-speed vortices that vorticity-based approaches consistently miss. We apply the difference form of SPH to compute the velocity gradient as

$$\nabla \mathbf{u}_i = \sum_{j \in \mathcal{N}_i^F} \frac{m_j}{\rho_j} (\mathbf{u}_j - \mathbf{u}_i) \otimes \nabla W(\mathbf{x}_i - \mathbf{x}_j), \tag{6}$$

where $\mathbf{a} \otimes \mathbf{b} = \mathbf{a}\mathbf{b}^T$, $m_j$ and $\rho_j$ denote particle mass and density, and $W$ refers to the SPH smoothing kernel. The global encoder abstracts scene-level vortical context through KVN-guided particle sampling followed by pooled multi-head attention (PMA) aggregation. Our approach employs a dynamic sampling strategy that continuously updates the sample set $\mathcal{S}$ based on evolving KVN distributions. Unlike static sampling schemes, this mechanism adapts to temporal changes in vortical structures by incorporating acceptance and rejection probabilities derived from particle importance. For each fluid particle $p_i$, we define an acceptance probability $\alpha_i$ and a rejection probability $\beta_i$ based on its KVN value as

$$\alpha_i = 1 - \frac{1}{[\kappa_i + 1]^a}, \quad \beta_i = e^{-b\kappa_i}, \tag{7}$$

where $a > 0$ and $b > 0$ are hyperparameters controlling sample generation and removal rates respectively. The acceptance probability increases monotonically with KVN, ensuring that particles exhibiting rigid-body-like rotation have higher chances of being selected. Conversely, the rejection probability decreases with KVN, allowing high-importance particles to remain in the sample set longer. At each time step, a uniform random number $\xi \in [0,1]$ is generated for each particle. For particles not currently in $\mathcal{S}$, inclusion occurs when $\xi < \alpha_i$. Conversely, existing samples are removed when $\xi < \beta_i$. This probabilistic mechanism ensures that high-KVN regions maintain dense sampling while allowing for natural sample turnover as vortical structures evolve.

The sampled particles are processed by the local encoder to obtain feature embeddings $\{\mathbf{h}_l^{(L)}\}_{l \in \mathcal{S}}$, where $\mathbf{h}_l^{(L)}$ denotes the final layer output from the local encoder for particle $p_l$. These local embeddings are concatenated with positional information to form enhanced representations $\mathbf{z}_l = [\mathbf{h}_l^{(L)}; \mathbf{x}_l]$ for each sampled particle. The embeddings are then aggregated into a global token $\mathbf{g}$ using pooled multi-head attention as

$$\mathbf{g} = \text{MHA}(\mathbf{q}, \{\mathbf{z}_l\}_{l \in \mathcal{S}}, \{\mathbf{z}_l\}_{l \in \mathcal{S}}), \tag{8}$$

where $\text{MHA}(\cdot)$ denotes multi-head attention and $\mathbf{q}$ is a learnable query vector that extracts the most relevant global information from the sampled particle set. Pre- and post-layer normalization ensures stable training and prevents gradient degradation during the attention pooling process.

## 4.3 DECODER WITH GLOBAL-LOCAL FUSION

The decoder integrates global vortical context with local particle embeddings to predict velocity corrections. The architecture employs a two-stage design with gated fusion mechanisms that balance global scene understanding with local particle dynamics. The first stage establishes global context by conditioning on both the global token $\mathbf{g}$ and query particle positions $\mathbf{x}_i$ as

$$\mathbf{f}_i^{(1)} = \mathcal{C}_1(\mathbf{W}_{\text{pos}}\mathbf{x}_i + \gamma_g \cdot \mathbf{W}_g \text{LN}(\mathbf{g}), \mathbf{g}), \tag{9}$$

where $\mathbf{W}_{\text{pos}}$ and $\mathbf{W}_g$ are learnable linear transformation matrices, $\text{LN}(\cdot)$ denotes layer normalization, $\gamma_g$ is a learnable gating parameter initialized to a small value to ensure stable training and prevent the global token from overwhelming local spatial information during early training stages, and $\mathcal{C}_1(\cdot, \mathbf{g})$ represents a sequence of conditional residual blocks that use the global token to modulate intermediate representations. The second stage performs the fusion between the global and local embeddings as

$$\mathbf{f}_i^{(\text{fused})} = \mathcal{C}_2(\text{GELU}(\mathbf{W}_{\text{fuse}}([\gamma_g \mathbf{f}_i^{(1)}; \gamma_l \mathbf{h}_i^{(L)}])), \mathbf{g}), \tag{10}$$

where $\mathbf{h}_i^{(L)}$ denotes the final layer output from the local encoder for particle $p_i$, $\mathbf{W}_{\text{fuse}}$ is a learnable linear transformation that processes the concatenated features, and $\mathcal{C}_2(\cdot, \mathbf{g})$ represents additional conditional residual blocks for refinement. The learnable gates $\gamma_g$ and $\gamma_l$ are trained to optimally balance the contributions from global context and local neighborhoods across the entire dataset. The conditional residual blocks further refine the fused representation. These blocks maintain the global conditioning throughout the refinement process, ensuring that scene-level vortical patterns continue to inform the local predictions. The final velocity correction incorporates both the refined fused representation and a direct skip connection from local features by

$$\delta\hat{\mathbf{u}}_i = \mathbf{W}_{\text{out}}\mathbf{f}_i^{(\text{fused})} + \gamma_{\text{skip}} \cdot \mathbf{W}_{\text{skip}}\mathbf{h}_i^{(L)}, \tag{11}$$

where $\mathbf{W}_{\text{out}}$ and $\mathbf{W}_{\text{skip}}$ are learnable output projection matrices. This residual design preserves local fluid dynamics while providing stable gradient flow, ensuring that fine-grained vortical details captured by the local encoder are maintained throughout the global-local fusion process.

## 5 EXPERIMENTS

We train the network end-to-end using a particle-wise mean squared error loss as

$$\mathcal{L} = \frac{1}{N}\sum_{i=1}^{N}\|\delta\hat{\mathbf{u}}_i - \delta\mathbf{u}_i\|_2^2, \tag{12}$$

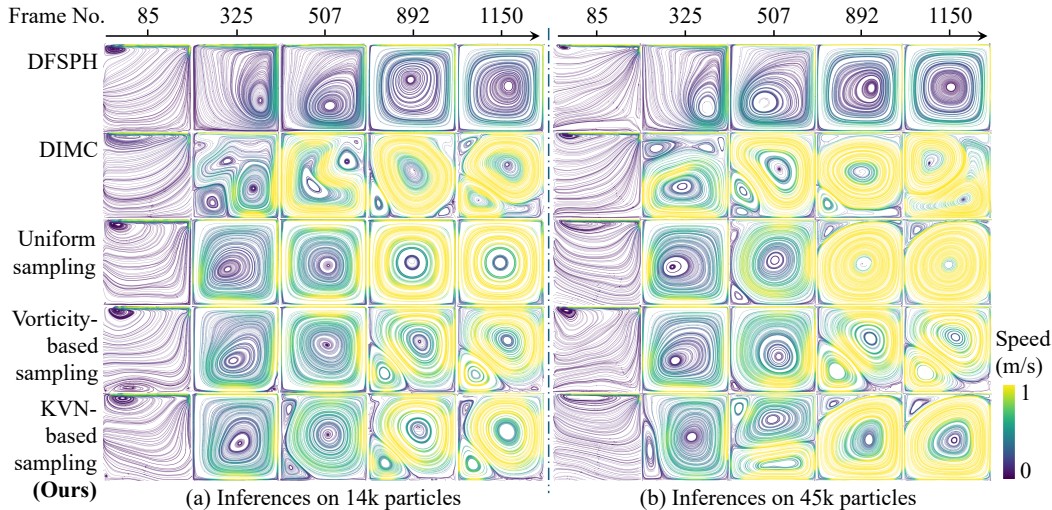

Figure 3: Lid driven cavity comparison across different particle resolutions and sampling methods. All learning-based methods use 5,000 samples and are trained on 14k particle DIMC simulations. (a) Inference on 14k particles: DFSPH shows no vortical structures, while uniform sampling produces a central vortex. Our KVN-based sampling generates secondary vortices earlier than vorticity-based sampling. (b) Inference on 45k particles: DFSPH exhibits weak vortical flows due to improved discretization. Vorticity-based sampling also shows delayed emergence of secondary vortices, while our KVN-based approach produces coherent vortical structures at both particle resolutions, demonstrating capability in identifying and reinforcing nascent low-angular-speed vortices.

where the target velocity corrections $\delta \mathbf{u}_i$ are computed using the Monte Carlo Biot-Savart law as described in Section 3. The loss is averaged per particle to ensure that optimization focuses on achieving accurate per-particle corrections.

**Implementation Details.** Our network architecture employs local encoder layers with hidden dimensions of [32, 64] and uses 8-head multi-head attention for global token aggregation. We optimize using AdamW with an initial learning rate of $10^{-3}$ and weight decay of $10^{-5}$, employing linear warmup followed by linear decay scheduling. Training is conducted with mixed precision (bfloat16) to improve computational efficiency. The decoder uses conditional residual blocks with layer-wise scaling for stable gradient flow, while learnable gating parameters balance global and local feature contributions during training.

**Experimental Setup.** We evaluate our angular-speed-invariant importance sampling approach across multiple SPH scenarios. Training data is generated using Dynamic Importance Monte Carlo (DIMC) method Ye et al. (2025), which provides high-quality vortex-enhanced SPH simulations as supervision targets. We use Divergence-Free SPH (DFSPH) Bender & Koschier (2017) as the baseline representing standard SPH simulation without vortex enhancement. For clarity, we use "Our Local" to refer to our continuous convolution-based local encoder, and "CConv" to refer to the continuous convolution method from Ummenhofer et al. Ummenhofer et al. (2020). To facilitate controlled comparisons across different aspects of our method, we employ varying configurations. For sampling strategy comparisons and sample count sensitivity analysis, we simplify the feature vector to $[1, \delta\boldsymbol{\omega}]$ (excluding velocity) and use fixed sample counts rather than dynamic sampling to ensure controlled comparison conditions. In the Kármán vortex street generalization experiments, we restore the full feature vector $[1, \mathbf{u}, \delta\boldsymbol{\omega}]$ and dynamic sampling to demonstrate the complete method's transferability. All experiments are conducted with varying particle counts and obstacle configurations to demonstrate robustness across different scenarios.

**Comparison of sampling strategies.** To evaluate the robustness of different sampling strategies across varying spatial discretizations, we conduct a lid driven cavity experiment with two particle resolutions. We train all learning-based methods using DIMC simulations with 14k particles, where

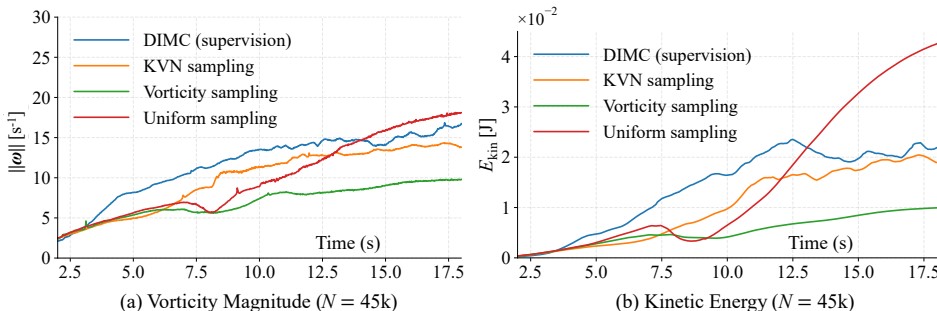

(a) Vorticity Magnitude ($N = 45$k)  (b) Kinetic Energy ($N = 45$k)

Figure 4: Quantitative evaluation of sampling methods on lid driven cavity with 45k particles. (a) Vorticity magnitude and (b) kinetic energy evolution over time. Both metrics show consistent patterns: KVN-based sampling closely tracks DIMC supervision, vorticity-based sampling plateaus at low levels, while uniform sampling exhibits late over-amplification for the central vortex.

each method samples 5,000 particles, then test on both 14k and 45k particle configurations while maintaining the same sample count.

Figure 3 shows streamline results for a $1 \times 1$ m box with the top lid driven at 1 m/s. At the coarser resolution of 14k particles, as shown in Fig. 3(a), DFSPH produces only a single central vortex with no secondary structures, while learning-based methods generate additional vortices with different timing patterns. Our KVN-based sampling produces secondary vortices earlier in the simulation compared to both uniform and vorticity-based approaches, demonstrating superior identification of nascent rotational structures. When the particle count increases to 45k, similar patterns emerge in Fig. 3(b): KVN-based sampling produces coherent vortical patterns across both resolutions, while vorticity-based sampling exhibits delayed vortex formation.

Figure 4 provides quantitative validation through vorticity magnitude and kinetic energy evolution over time, both exhibiting consistent patterns across sampling strategies. KVN-based sampling maintains levels that closely track the DIMC supervision for both metrics. Vorticity-based sampling plateaus at significantly lower values, confirming its limited ability to sustain vortical structures. Uniform sampling presents an interesting contrast: while it eventually achieves the highest values, this occurs through excessive amplification in the latter half of the simulation, corresponding to over-enhancement of the dominant central vortex rather than balanced development of multiple structures. These results highlight the key advantage of angular-speed-invariant importance sampling. Since vorticity scales with angular velocity, vorticity-based methods overlook nascent vortices with low rotational speeds, leading to delayed reinforcement. KVN provides an angular-speed-invariant measure that equally emphasizes rotational cores regardless of current velocity, enabling identification and strengthening of small, slowly rotating structures that would otherwise dissipate. The quantitative metrics confirm that effective vortex enhancement requires balanced preservation rather than simple amplification.

**Sample count sensitivity analysis.** To investigate sample count sensitivity, we conduct a rotating panel experiment with 14k particles where a square panel rotates at $1.5\pi$ rad/s. Figure 5 compares DFSPH baseline, DIMC supervision, and our method using sample counts from 500 to 8000. Our method demonstrates consistent vortex patterns across all sample counts as shown in Fig. 5(c), with generally stronger vortex reinforcement at higher sample counts. The quantitative analysis in Fig. 5(d) shows that average vorticity magnitude exhibits an overall upward trend with increasing sample count, though with some fluctuations. Based on visual comparison of streamline graphs with the DIMC target, optimal performance occurs at 4000-6000 samples (28.6-42.9% of total particles) where the vortex patterns most closely resemble the supervision data. At 8000 samples (57.1% of particles), over-reinforcement becomes evident with excessively strong vortical flows, indicating an optimal sampling ratio of approximately 30-40%. These results demonstrate that our learning framework with KVN-based sampling provides robust vortex enhancement across a wide range of sample densities while revealing clear performance optima.

The computational efficiency comparison reveals a key advantage of learning-based approaches over physics-based methods. While our model's inference time remains relatively stable across

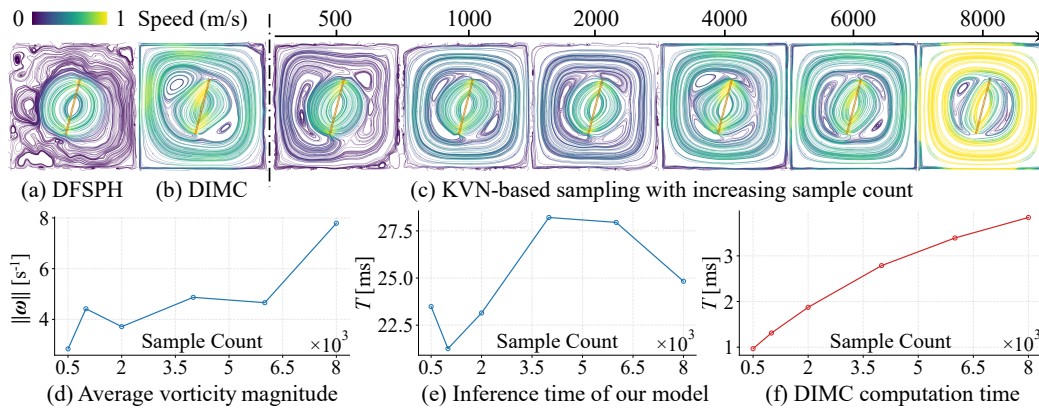

Figure 5: Sample count sensitivity analysis on rotating panel scenario (14k particles, $1.5\pi$ rad/s). (a) DFSPH baseline. (b) DIMC supervision. (c) Our method with sample counts 500-8000, showing consistent patterns with optimal performance at 4000-6000 samples. (d) Average vorticity magnitude increases with sample count. (e) Our model inference time remains stable. (f) DIMC computation time scales significantly with sample count, indicating scalability advantages of learning-based approaches.

different sample counts, as demonstrated in Fig. 5(e), DIMC's computation time scales dramatically with sample counts, shown in Fig. 5(f), increasing over 3-fold from 500 to 8000 samples. This suggests that learning-based methods will demonstrate significant computational advantages over physics-based sampling approaches as simulation scales further increase. This analysis confirms that our learning framework effectively integrates KVN-based importance sampling to concentrate computational resources on physically relevant regions while providing scalability advantages over Monte Carlo methods.

**Evaluation on unseen configurations.** To assess how well our angular-speed-invariant sampling transfers to different obstacle arrangements, we train our model on DIMC supervisory data from a simple sphere scenario and evaluate on Kármán vortex street configurations with varying multi-obstacle layouts. Figure 6 shows that while these configurations were not seen during training, our method maintains effective vortex enhancement across all test cases. The DFSPH baseline Fig. 6(a) produces minimal vortical structures due to numerical dissipation, regardless of obstacle configuration. The DIMC target Fig. 6(b) demonstrates the desired vortex patterns that should form behind each obstacle arrangement. Our method Fig. 6(c) successfully reproduces similar vortical structures across all four unseen configurations, with vortex formation and evolution patterns that closely match the DIMC targets.

Table 1 provides quantitative analysis of different architectural combinations with established methods including PointNet++ (Qi et al. (2017)), Point Transformer (Zhao et al. (2021)), and Continuous Convolutions (CConv) (Ummenhofer et al. (2020)). An interesting observation is that all methods without KVN Global already achieve reasonably good performance with minimal qualitative differences, suggesting that the local network input features $[1, \mathbf{u}, \delta\boldsymbol{\omega}]$ provide saturated representation for the current vortex enhancement scenario. However, KVN Global consistently extracts additional performance gains from this saturated baseline across all architectures (PointNet++: +0.9% average vorticity, Point Transformer: +0.9% average vorticity, CConv: +0.5% average vorticity). The fact that all methods show positive improvements indicates these gains are systematic rather than due to experimental noise, demonstrating KVN Global's effectiveness in enhancing vortical structure learning even when input features approach their representational limits.

## 6 CONCLUSION

We introduce angular-speed-invariant importance sampling guided by the Kinematic Vorticity Number (KVN) to improve learning-based vortex enhancement in SPH fluids. Unlike vorticity magnitude, which scales with angular speed and biases models toward fast-spinning structures, KVN

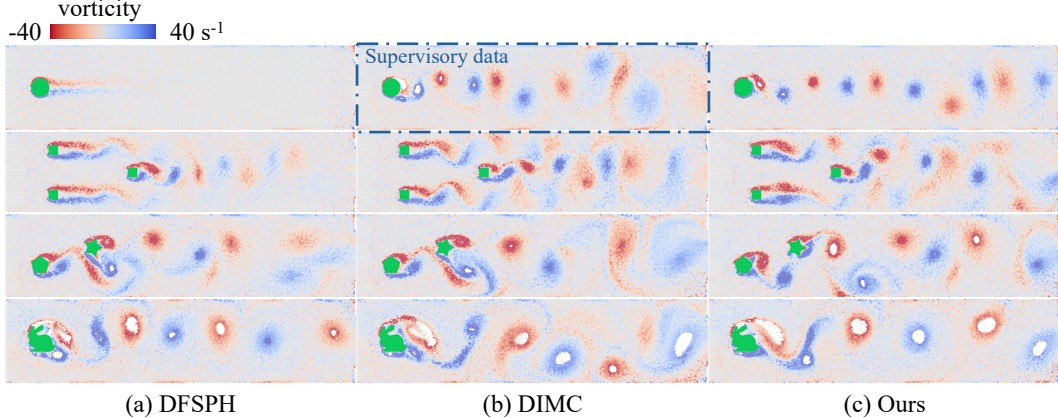

Figure 6: Evaluation on unseen obstacle configurations in Kármán vortex street scenarios. Our model is trained on DIMC supervisory data from a simple sphere scenario (highlighted by dashed box) but evaluated on four different multi-obstacle configurations. (a) DFSPH baseline shows limited vortical structures. (b) DIMC target demonstrates rich vortex patterns behind obstacles. (c) Our angular-speed-invariant sampling approach produces coherent vortex structures across all unseen configurations, demonstrating effective transfer of rotational quality identification principles.

Table 1: Quantitative ablation study and architectural comparison on Von Kármán vortex street scenario. All baseline methods achieve saturated performance with input features $[1, \mathbf{u}, \delta\boldsymbol{\omega}]$, yet KVN Global consistently extracts additional gains across different encoders, demonstrating its effectiveness even at representational limits.

| Method | Max $\|\boldsymbol{\omega}\|$ | Avg. $\|\boldsymbol{\omega}\|$ | Avg. $E_{\mathrm{kin}}$ | Avg. time (ms) |
|---|---|---|---|---|
| **Ours (KVN global on our local)** | 13.368 | 9.839 | 0.131 | 31.5 |
| **Our local only with KVN feature** | 13.602 | 9.620 | 0.126 | 24.3 |
| **Our local only** | 13.141 | 9.348 | 0.126 | 26.9 |
| CConv with KVN Global | 12.807 | 9.744 | 0.130 | 38.2 |
| CConv | 13.574 | 9.693 | 0.128 | 35.7 |
| Point Transformer with KVN Global | 14.460 | 9.811 | 0.132 | 127.4 |
| Point Transformer | 13.571 | 9.726 | 0.130 | 128.4 |
| PointNet++ with KVN Global | 14.432 | 10.006 | 0.133 | 53.3 |
| PointNet++ | 12.573 | 9.918 | 0.132 | 51.4 |

provides a dimensionless measure that equally emphasizes rotational cores regardless of their current velocity. This property enables reliable identification and reinforcement of nascent low-speed vortices that conventional vorticity-based and uniform sampling methods consistently overlook.

Our experimental evaluation across multiple SPH scenarios confirms the effectiveness of this approach. In lid-driven cavity experiments, KVN-guided sampling produces secondary vortices earlier than competing methods while maintaining physically consistent energy evolution. Sample count sensitivity analysis reveals optimal performance at 30-40% of total particles. Moreover, our method exhibits scalability potential. While Monte Carlo methods like DIMC show dramatic computational cost increases with sample count (over 3-fold from 500 to 8000 samples), our learning-based approach maintains stable inference times as sample count increases. The architectural comparison experiments validate that KVN-derived global context provides consistent improvements across different local encoders. Notably, all baseline methods achieve saturated performance with local features $[1, \mathbf{u}, \delta\boldsymbol{\omega}]$, yet KVN Global consistently extracts additional gains across architectures. This systematic enhancement from saturated baselines demonstrates KVN Global's effectiveness even at representational limits, where modest improvements carry significant meaning.

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

## A USE OF LLMS

We employed Large Language Models (LLMs) in guidance when implementing baseline methods, English grammar or style edits, as well as assistance in locating and organizing related literature.

# B  CONSTRUCTING THE VELOCITY CORRECTION TARGET

This section describes the construction of velocity correction targets $\delta\mathbf{u}_i$ used for model supervision. Our approach follows the standard vorticity-loss methodology in SPH vortex enhancement.

## B.1  VORTICITY LOSS FRAMEWORK

We begin with the vorticity transport equation derived from the curl of the Navier-Stokes equations as

$$\frac{D\boldsymbol{\omega}}{Dt} = (\boldsymbol{\omega} \cdot \nabla)\mathbf{u} + \nu\nabla^2\boldsymbol{\omega} + \nabla \times \mathbf{f}, \tag{13}$$

where $\boldsymbol{\omega} = \nabla \times \mathbf{u}$ is the vorticity field.

At each simulation step, we compute the current vorticity $\boldsymbol{\omega}^n$ and evolve it to obtain $\boldsymbol{\omega}^{n+1}$. Simultaneously, we apply non-pressure forces to the velocity field to get intermediate velocity $\bar{\mathbf{u}}^n$. The vorticity loss quantifies the dissipated rotational structures by

$$\delta\boldsymbol{\omega}^n = \boldsymbol{\omega}^{n+1} - \nabla \times \bar{\mathbf{u}}^n. \tag{14}$$

## B.2  SPH DISCRETIZATION

We discretize differential operators using SPH's difference formulation. For particle $i$ with neighbors $\mathcal{N}_i$,

$$(\nabla \times \mathbf{u})_i = \sum_{j \in \mathcal{N}_i} \frac{m_j}{\rho_j}(\mathbf{u}_j - \mathbf{u}_i) \times \nabla W(\mathbf{x}_i - \mathbf{x}_j), \tag{15}$$

$$\nabla\mathbf{u}_i = \sum_{j \in \mathcal{N}_i} \frac{m_j}{\rho_j}(\mathbf{u}_j - \mathbf{u}_i) \otimes \nabla W(\mathbf{x}_i - \mathbf{x}_j), \tag{16}$$

where $W$ is the SPH kernel function and $\mathbf{a} \otimes \mathbf{b} = \mathbf{a}\mathbf{b}^T$.

## B.3  VELOCITY TARGET GENERATION

To convert vorticity loss into velocity corrections, we need to solve the velocity-vorticity Poisson equation as

$$-\nabla^2\mathbf{u} = \nabla \times \boldsymbol{\omega}. \tag{17}$$

Using Green's function for infinite domains, the Biot-Savart solution is

$$\delta\mathbf{u}_i = v\sum_{j \in \mathcal{H}} \nabla G(\mathbf{x}_i - \mathbf{x}_j) \times \delta\boldsymbol{\omega}_j^n, \tag{18}$$

where $G(\mathbf{r}) = \frac{1}{2\pi}\ln\|\mathbf{r}\|$ (2D) or $G(\mathbf{r}) = \frac{1}{4\pi\|\mathbf{r}\|}$ (3D), $v$ is the vortex particle volume, and $\mathcal{H}$ contains all fluid particles.

For training data generation, we use Monte Carlo estimation with importance sampling as

$$\delta\mathbf{u}_i = \frac{v}{|\mathcal{S}|}\sum_{j \in \mathcal{S}} \frac{\nabla G(\mathbf{x}_i - \mathbf{x}_j) \times \delta\boldsymbol{\omega}_j^n}{P_j}, \tag{19}$$

where $\mathcal{S}$ is a sampled subset and $P_j$ is the sampling probability of particle $j$.

