# OpenReview forum: "Learning Vortex Enhancement with Angular-Speed-Invariant Importance Sampling in SPH Fluids"
_ICLR.cc/2026/Conference — ICLR 2026 Conference Withdrawn Submission_

### Official Review · Reviewer_5X6E · 2025-11-01

**Soundness:** 3
**Presentation:** 2
**Contribution:** 2
**Rating:** 2
**Confidence:** 4

**Summary:**

This paper presents a learning-based framework for vortex enhancement in Smoothed Particle Hydrodynamics (SPH) fluids using angular-speed-invariant importance sampling. A particle-based neural network is proposed to improve energy (vorticity) conservation in Smoothed Particle Hydrodynamics flow solvers. The authors propose a particle resampling strategy based on the Kinematic Vorticity Number (KVN), which measures the degree of rigid-body rotation independently of angular velocity. Particles with high KVN values are pooled via attention into a global token that encodes vortical context. The network is trained to predict velocity corrections derived from a Biot–Savart mapping of the vorticity loss field. The authors show that integrating KVN-based features benefits different encoder architectures; the main application of the proposed approach is to recover vorticty highlighting the generality of angular-speed-invariant sampling for learning vortex enhancement in SPH.

**Strengths:**

- Employing an angular-speed-invariant importance sampling metric is a clear and well-motivated contribution, offering a physically inspired sampling strategy improvement over vorticity-magnitude-based sampling.

- The combination of KVN-guided sampling with continuous convolutions and global attention is conceptually elegant and well-integrated within the learning framework.

**Weaknesses:**

- The rationale behind energy dissipation in SPH seems is not thoroughly discussed. The authors blame kernel approximations and coarse discretizations, but there are more to it such as NS variables choice (vorticity vs velocity-based vs impulse-based), hybrid simulation discretizations (MPM, FLIP), accuracy of pressure projection solver, etc. Solving this issue with a neural approach without understanding the root cause seems to be a sub-par choice that could lead just to additional overheads without a clear practical advantage.

- The formulation of eq, (1) is not clear. Does the vorticity loss term represents the difference between consecutive steps or a cumulative dissipation measure?

- The authors propose a neural approach in which the potential benefit would be to increase the efficiency of SPH solvers. The issue however is that the paper does not provide detailed timing comparisons against simply increasing the particle count of a baseline solver, leaving the theoretical computational efficiency advantage unverified.

- The evaluation focuses primarily on 2D configurations, such as lid-driven cavity and rotating panel setups, which limits the assessment of generality to more complex or 3D flow regimes.

- While the results qualitatively demonstrate earlier vortex formation, quantitative metrics beyond vorticity magnitude and kinetic energy would strengthen the validation (e.g., divergence error, spectral energy distribution).

**Questions:**

- Did you try enhancing the solid particles features with the obstacles normal information?
- Could the strain tensor on equation 6 be zero? If yes, a delta would have to be added to the equation to avoid instabilities.
- The multi-head attention is similar to self-attention correct? The text would be clearer if this is explicitly stated. Also would be interesting to see how does the proposed approach scales with the number of particles.

---

> ### Author Response · Authors · 2025-11-13
>
> We appreciate the reviewer’s constructive insights.
> In contrast to Eulerian methods, where advection is often identified as the primary cause of dissipation, SPH experiences dissipation from several interacting sources. These include kernel approximation errors, integration schemes, and the accuracy of pressure projection. Because these factors influence one another, it is difficult to determine a single dominant mechanism of dissipation. Considering this complexity and the computational cost required to address each source individually, a compensation-based strategy is often more practical. Our intention in this work is therefore to investigate whether a neural correction can mitigate the dissipation and preserve vortices in a cost-effective manner.
> We also acknowledge that the claimed efficiency benefits were not rigorously validated, as timing comparisons against higher-resolution SPH baselines are missing. In addition, the evaluation is limited to 2D configurations, and more comprehensive 3D experiments and physical metrics are needed to assess generality. These broader issues will be addressed in a substantially revised future version of the work.

---

### Official Review · Reviewer_4cAh · 2025-11-07

**Soundness:** 2
**Presentation:** 2
**Contribution:** 2
**Rating:** 2
**Confidence:** 3

**Summary:**

This paper proposes a framework for enhancing vortex structures in Smoothed Particle Hydrodynamics (SPH) fluid simulations through a physically grounded sampling mechanism called angular-speed-invariant importance sampling. The key idea is to employ the Kinematic Vorticity Number (KVN) as an importance measure to select particles that contribute most significantly to rotational motion, thereby overcoming the bias of traditional vorticity-magnitude–based approaches toward high-speed vortices. The method combines Continuous Convolution (CConv) layers for local neighborhood encoding and a KVN-guided attention pooling mechanism for capturing global flow context. The learning objective is derived from the Biot–Savart law, enabling the network to predict velocity corrections that improve vortex coherence. Experiments on multiple canonical SPH test cases are conducted.

**Strengths:**

The use of KVN represents a meaningful and well-motivated improvement to the data sampling process in SPH-based simulations.

**Weaknesses:**

1. The focus of the paper is narrow and largely application-centric, limiting its relevance to a general machine learning audience. While the work is well-executed within SPH-based vortex modeling, it lacks a clear articulation of how the proposed ideas advance the broader field of ML or operator learning, which is crucial for ICLR.

2. The writing and narrative structure of the paper can be significantly improved. The flow of ideas sometimes feels fragmented with abrupt transitions. The motivation of this work to the ICLR community can be much better articulated and substantiated by improving the quality of writing of this work.

3. The machine learning contribution is not sufficiently emphasized. Most of the novelty lies in the sampling metric (KVN) rather than in the learning framework itself. The method does not introduce new learning paradigms, architectures, or optimization techniques that generalize beyond this domain.

4. The discussion of theoretical properties of the KVN-based sampling approach is minimal. There is no quantitative analysis of variance reduction, sampling efficiency, or convergence characteristics that would substantiate the claimed advantages of angular-speed invariance.

5. The quantitative improvements reported are relatively small and often confined to specific metrics. While qualitative visualizations are convincing, it is unclear whether the observed gains are statistically significant or impactful in practical SPH workflows.

6. The paper does not compare against other relevant ML-based baselines relevant to fluid simulation, such as Graph Neural Networks, Physics-Informed Neural Networks (PINNs), or differentiable SPH solvers (e.g., Neural SPH). Without such comparisons, it is difficult to assess the true performance or generalization advantage of the proposed method.

**Questions:**

1. How does the proposed KVN-based sampling approach generalize to three-dimensional SPH flows or higher-Reynolds-number turbulence, where vortex interactions are more complex?

2. How does this method compare to state-of-the-art ML-based fluid simulation frameworks, including Graph Neural Networks for SPH or differentiable physics models such as Neural SPH and Neural Vortex Method?

3. Can the authors provide more insight into how the proposed KVN-based sampling framework might generalize to other physical domains such as AI for plasma dynamics, meteorology, or materials modeling to establish broader relevance for the ICLR community?

4. How sensitive are the results to the hyperparameters used in defining KVN or in the model training process, and how do these affect reproducibility or model stability?

---

> ### Author Response · Authors · 2025-11-13
>
> We acknowledge that the deep learning contribution is underdeveloped in the current submission. While the KVN-sampling strategy is presented only within the context of vortex enhancement, the underlying idea that extract pure rotation features independent of angular speed could potentially benefit a broader class of problems where rotational structures are relevant. Nevertheless, we did not sufficiently articulate this generality in the present version. We appreciate the reviewer’s constructive perspective, which will meaningfully inform the next version of the work.

---

### Official Review · Reviewer_2ew8 · 2025-11-07

**Soundness:** 2
**Presentation:** 2
**Contribution:** 3
**Rating:** 4
**Confidence:** 3

**Summary:**

This paper studies how to improve vortex detection and simulation. While previous baselines make use of the vorticity magnitude, this method proposes to use Kinematic Vorticity Number instead. After introducing the  KVN, the paper describes the chosen architecture and proposes some experiment to illustrate the improvements when using KVN.

**Strengths:**

- The experimental results showcase an improvement wrt to baseline.
- The proposed method introduces a new component that was not used in the littérature, to the best of my knowledge.

**Weaknesses:**

-	I found the paper a little hard to read. It would benefit of a more fluent wrirting (especially introduction), to clearly introduce each component before reading the paper. This is even more important in general conferences such as ICLR, where people from different horizons could read this work. I am not an expert in particle fluid dynamic, however, I know well deep learning and its application to physical systems.
-	The scope of applicability seems limited to a specific type of physical systems.
-	While some experiments are convincing, I think the experimental part could be improved to be even more convincing on some aspects, see questions.
-	Missing references : a huge litterature on mesh-free method have emerged recently. I think a paragraph to position the current work wrt these methods should be included (could be in appendices). These methods use eg either INR [1-2], attention [3-6], graphs [7-9] and many others. Moreover, since continuous convolutions are a core component of the work, I think additional related work should appear.  Moreover, a comparison with some of these method would allow the reader to understand the benefit of the proposed method wrt to others.

[1] Operator Learning with Neural Fields: Tackling PDEs on General Geometries, Louis Serrano, Lise Le Boudec, Armand Kassaï Koupaï, Thomas X Wang, 2023.

[2] Implicit Neural Spatial Representations for Time-dependent PDEs, Honglin Chen, Rundi Wu, Eitan Grinspun, Changxi Zheng, Peter Yichen Chen, 2023

[3] Transolver: A Fast Transformer Solver for PDEs on General Geometries, Haixu Wu, Huakun Luo, Haowen Wang, Jianmin Wang, Mingsheng Long, 2024.

[4] Universal Physics Transformers: A Framework For Efficiently Scaling Neural Operators, Benedikt Alkin, Andreas Fürst, Simon Schmid, Lukas Gruber, Markus Holzleitner, Johannes Brandstetter, 2024

[5] AROMA: Preserving Spatial Structure for Latent PDE Modeling with Local Neural Fields, Louis Serrano, Thomas X Wang, Etienne Le Naour, Jean-Noël Vittaut, Patrick Gallinari, 2024

[6] CViT: Continuous Vision Transformer for Operator Learning, Sifan Wang, Jacob H Seidman, Shyam Sankaran, Hanwen Wang, George J. Pappas, Paris Perdikaris, 2024.

[7] Learning Mesh-Based Simulation with Graph Networks, Tobias Pfaff, Meire Fortunato, Alvaro Sanchez-Gonzalez, Peter W. Battaglia, 2020

[8] Geometry-Informed Neural Operator for Large-Scale 3D PDEs, Zongyi Li, Nikola Borislavov Kovachki, Chris Choy, Boyi Li, Jean Kossaifi, Shourya Prakash Otta, Mohammad Amin Nabian, Maximilian Stadler, Christian Hundt, Kamyar Azizzadenesheli, Anima Anandkumar, 2023.

[9] RIGNO: A Graph-based framework for robust and accurate operator learning for PDEs on arbitrary domains, Sepehr Mousavi, Shizheng Wen, Levi Lingsch, Maximilian Herde, Bogdan Raonić, Siddhartha Mishra, 2025.

**Questions:**

### Presentation
-	Could you add details on the positioning of the paper? ie why is it so important to improve vortex detection/simulation ? I know it has a lot of industrial applications, but this will help understant the importance and use case of such methods.
-	The experiment figure 2 is a very good illustration and motivating example that could be used to motivate the method at earlier stages in the paper. Moreover I think it would benefit of being more discussed : what are the expected behavior ? Why are the cases a and b wrong ?

### Clarity
-	I think that the main figure 1, is a bit hard to read. Maybe highlighting arrows/link between blocks would help crealy identifying the role of each block.
-	Line 221-223 : what happend if $ \xi > \alpha $ and $\beta$ ?
-	The architecture is not discussed in the experiment, while being described for several paragraphs. This blurs the message, which is focused on the new KVN feature.

### Experiment :
-	Ablation on $\gamma_g$, what removing it would do? It is mentioned line 244 that it ensure stable training. I think a small experiment to illustrate this sentence will strengthen this choice. Why are there 2 $\gamma_g$ (line 241 and 249)?
-	It is unclear for me, what is the ground truth/expected behavior of method, is it the DIMC method ? Then, while KVN seems to be far from the gt simulations on some example (eg fig 3b)?
-	In figure 3, what is the network architecture used for each method?
-	What do you mean by inference on 45k particles ? are all point treated with one forward pass of the neural network ? This represents a lot of points, making the method scalable, but this is not discussed. How does scales the method wrt mesh size? in terms of memory consumption/inference time.... ?
-	Do you have experiment on public/standard dataset of the litterature ? This would help positionning the work with other references. I think datasets such as cylinder flow from [7] above contains vortexes, or some vorticity/smoke equations (eg from the well).
-	I think more recent baselines could be used to make the proposed comparison with existing method more convincing. As is, the most recent model is from 2021. There have been a huge littérature on this subject in the past few years (see eg reference in the weakness section).
- very Little experimental details are proposed. What architecture are used ? It is important to be as precise as possible to allow reproductibility of the results.

### Additional Questions
-	Fig 4, Do you have any insight about why the vorticity magnitude strongly increase for the uniform sampling ? It is straight and start at about 8 for both experiment.
-	As stated in the paper, The improvement brought by the KVN is minimal on several architecture. Doesn’t it mean that the greater wortex modelization would come from another aspect of the model ? Is sometimes even worsen the performances table 1). Wouldn't using more recent and advanced neural architecture improve the detection of vortexes?

### Minor comment
-	line 069 : Citation issue (Ma et. Al)

---

> ### Author Response · Authors · 2025-11-13
>
> We sincerely thank the reviewer for the detailed comments.
> 1. Positioning of this paper.
> In graphics-oriented fluid simulation, there is an inherent trade-off between efficiency and accuracy. Increasing resolution (finer particles or grids), reducing time steps, or employing higher-order numerical integrators (e.g., Runge–Kutta) can preserve high-frequency features such as vortices and splashes, but at a significant computational cost. Achieving both efficiency and visual fidelity remains a central goal in the graphics community.
> Standard SPH often loses high-frequency details for several reasons, including repeated interpolation, the use of forward Euler integration, and insufficient particle density. Although DIMC methods provide effective vorticity preservation with reasonable overhead, they still require a large number of vortex samples. Because the Biot–Savart kernel has non-compact support, each particle must query all samples during velocity correction, which becomes a bottleneck for large-scale scenarios.
> Our work aims to alleviate this bottleneck by proposing a neural approach that enhances vortical flow in SPH using a joint global token, thus avoiding per-particle queries over all samples. We do not claim that our network surpasses prior simulation methods; instead, we attempt to demonstrate how KVN-based global features can facilitate the training of a vortex-enhancing model.
> 2. Regarding Figure 2.
> All images in Figure 2, including (a) and (b), are correct. The figure illustrates why many prior vorticity-based approaches struggle to generate new vortices [1]. Vorticity highlights only regions of high angular velocity, while rotations with low angular speed are largely ignored. In contrast, the kinematic vorticity number identifies rotational behavior of any non-zero angular speed, providing a potential mechanism to address this limitation.
> [1] Smoothed Particle Hydrodynamics Techniques for the Physics-Based Simulation of Fluids and Solids. Koschier et al., 2019.
> We acknowledge that this reference may appear dated from an AI perspective, but it remains highly relevant within the graphics fluid-simulation literature, particularly for the specific topic addressed in this work.
>
> We will take these insights into account when substantially revising the work.

---

### Note · Authors · 2025-11-13

I have read and agree with the venue's withdrawal policy on behalf of myself and my co-authors.